# Quantifying normal and parkinsonian gait features from home movies: Practical application of a deep learning–based 2D pose estimator

**Kenichiro Sato\*, Yu Nagashima, Tatsuo Mano, Atsushi Iwata◉\*, Tatsushi Toda**

Department of Neurology, Graduate School of Medicine, University of Tokyo, Hongo, Bunkyo-ku, Tokyo, Japan

\* kenisatou-tky@umin.ac.jp (KS); iwata@m.u-tokyo.ac.jp (AI)

**Data Availability Statement:** The main data used in this study is distributed from CASIA database (http://www.cbsr.ia.ac.cn/english/Gait%20Databases.asp), which is not owned or collected

## Abstract

### Objective

Gait movies recorded in daily clinical practice are usually not filmed with specific devices, which prevents neurologists benefitting from leveraging gait analysis technologies. Here we propose a novel unsupervised approach to quantifying gait features and to extract cadence from normal and parkinsonian gait movies recorded with a home video camera by applying OpenPose, a deep learning–based 2D-pose estimator that can obtain joint coordinates from pictures or videos recorded with a monocular camera.

### Methods

Our proposed method consisted of two distinct phases: obtaining sequential gait features from movies by extracting body joint coordinates with OpenPose; and estimating cadence of periodic gait steps from the sequential gait features using the short-time pitch detection approach.

### Results

The cadence estimation of gait in its coronal plane (frontally viewed gait) as is frequently filmed in the daily clinical setting was successfully conducted in normal gait movies using the short-time autocorrelation function (ST-ACF). In cases of parkinsonian gait with prominent freezing of gait and involuntary oscillations, using ACF-based statistical distance metrics, we quantified the periodicity of each gait sequence; this metric clearly corresponded with the subjects' baseline disease statuses.

### Conclusion

The proposed method allows us to analyze gait movies that have been underutilized to date in a completely data-driven manner, and might broaden the range of movies for which gait analyses can be conducted.

by the authors. The authors did not receive special access privileges to the data.

**Funding:** The authors received no specific funding for this work.

**Competing interests:** The authors have declared that no competing interests exist.

# Introduction

Gait abnormality is one of the chief abnormal findings in chronic neurological diseases such as Parkinson's disease (PD); therefore, it is important for physicians and neurologists to examine parkinsonian gait, which is characterized by small steps, a reduced walking speed, insufficient foot elevation/progression [1,2], a forward-bent posture, freezing of gait (FOG), and festination [3] in the diagnosis and follow-up of PD.

In the daily clinical inpatient or outpatient setting, a gait evaluation consists of qualitative or semiquantitative assessment by neurologists or is based on patients' subjective complaints but not on quantified gait parameters such as cadence (= steps per minute) [4,5], step length [2], or foot clearance [1]. This is because a quantitative gait analysis requires specific devices such as a 3D motion capture system, accelerometer, or force plate [6,7], which are time-, labor-, space-, and cost-consuming to use daily. Moreover, while clinicians may routinely film their patients' gait in daily clinical settings (e.g. crowded and straightforward hallway in the hospital) and use a home video for later neurological assessment, these recorded clinical 2D movies without any annotations are practically unavailable for gait analyses. Thus, our daily clinical practice does not currently benefit from gait analysis technologies: the lack of quantitative metrics can eventually result in the potential bias of inter-patient comparison of gait with minor differences or atypical complex underlying pathophysiology.

To overcome these bottlenecks, it would be helpful if we could reliably quantify gait parameters without the need for specific devices. Here we propose use of OpenPose [8,9], a deep learning–based 2D-keypoint estimator that estimates the joint coordinates of persons in the pictures or videos obtained using a monocular camera, as it does not require external scales or markers. Using this estimator, we can automatically obtain the joint coordinates of persons in each picture/movie recorded, e.g. in a hospital hallway by a home video camera without specific equipment, thereby enabling the calculation of joint angles or other spatial parameters as processed in a conventional 3D motion capture system.

Here we aimed to investigate a method to quantify gait features in a data-driven manner, especially focusing on gait cadence, using the OpenPose application onto the daily clinical movies recorded from the frontal angle. First, we used OpenPose to convert normal gait movies to sequential joint coordinate data. Then, using the short-time autocorrelation function (ACF) [10,11], an applied form of a short-time Fourier Transform (STFT) [12,13], we conducted gait frequency detection. Next, we applied the same processing scheme to the pathological gait movies of our PD patient with prominent FOG. Since effective gait steps cannot always be easily determined for patients with significant FOG, especially those in movies recorded by a home video camera in which no accurate step annotation is available, we used the statistical distance–based approach that examined the timing at which temporal gait was relatively close to a normal one. Such an approach allows us to quantify the periodic step distribution during the gait sequences and obtain simple metrics to quantify the periodicity of each sequence. Our attempts might be the first step in shedding light on monocular home videos to quantify gait.

# Methods

## Sample data and preprocessing

This study was approved by the University of Tokyo Graduate School of Medicine institutional ethics committee (ID: 2339-(3)). The overall data processing flow is shown in Fig 1. The movie data used and the patient's demographics are summarized in Table 1. For video data of normal control gait, we obtained movies from the database provided by the Institute of Automation,

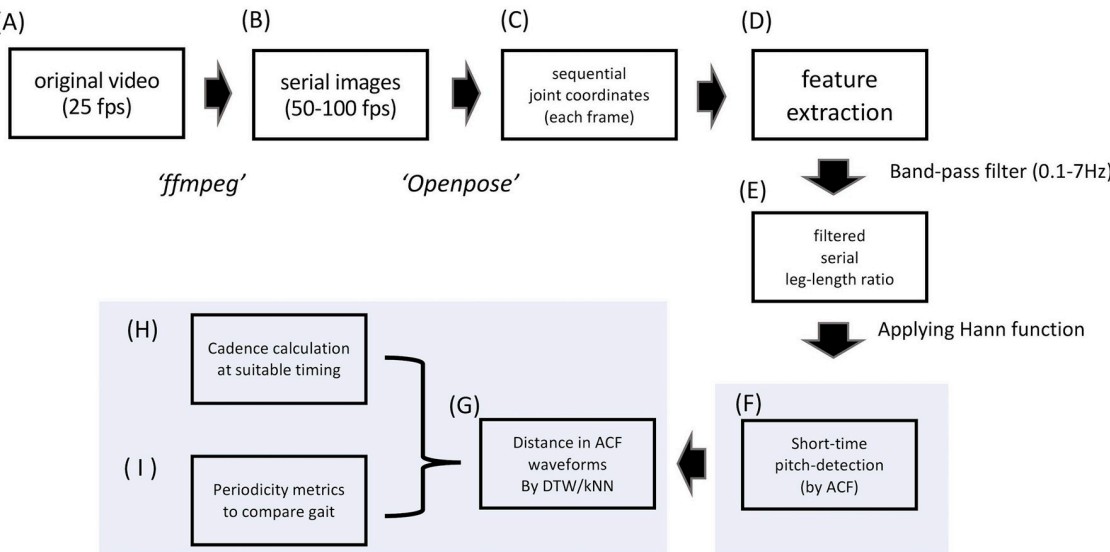

**Fig 1. Data processing flow.** Our proposed method consists of two distinct steps: upper line (A to E), deriving sequential gait feature data from movies via body joint coordinate extraction using OpenPose; lower line, estimating cadence from the sequential data using the short-time autocorrelation function and the subsequent analysis (F to I).

Chinese Academy of Sciences (CASIA Dataset-B) under permission [14]. We also obtained pathological gait videos from PD patients who were admitted to our hospital.

The CASIA database videos (Dataset-B) are videos of participants' walking recorded with a frame size of 320 × 240 and a frame rate of 25 frames per second (fps). Each participant has one gait sequence of 2–4 gait cycles recorded from the laterally viewed angle (i.e. seeing sagittal plane of gait) (Fig 2A-1) and frontally viewed angle (i.e. coronal plane of gait) (Fig 2B-1). Since the original video frequency was 25 fps, these videos were converted to serial *png* images at a sampling frequency of 50 or more (= 100 fps by default) [13] by using *ffmpeg* software [15] (Fig 1B).

**Table 1. Video data and subjects' characteristics.**

| data source | CASIA dataset-B | our hospital |
|---|---|---|
| participants | n = 117 | n = 2 (Pt.1 & Pt.2) |
| disease status | healthy control | Pt.1: PD Yahr 3 |
| | | Pt.2: PD Yahr 4 |
| recorded plane of gait | sagittal (laterally viewed) | frontal (frontally viewed) |
| | frontal (frontally viewed) | |
| distance between camera and the gait start point | not available | approximately 5 m |
| used video model | Fametec 318SC | Everio GZ-HD40 |
| movie fps | 25 | 30 |
| sequence used from each participant | n = 1 | Pt.1: n = 2 (before and after DBS) |
| | | Pt.2: n = 3 |
| age of participants | mostly in their 20's | Pt.1: 60's |
| | | Pt.2: 70's |
| sex of participants | male:female = 2:1 | both female |

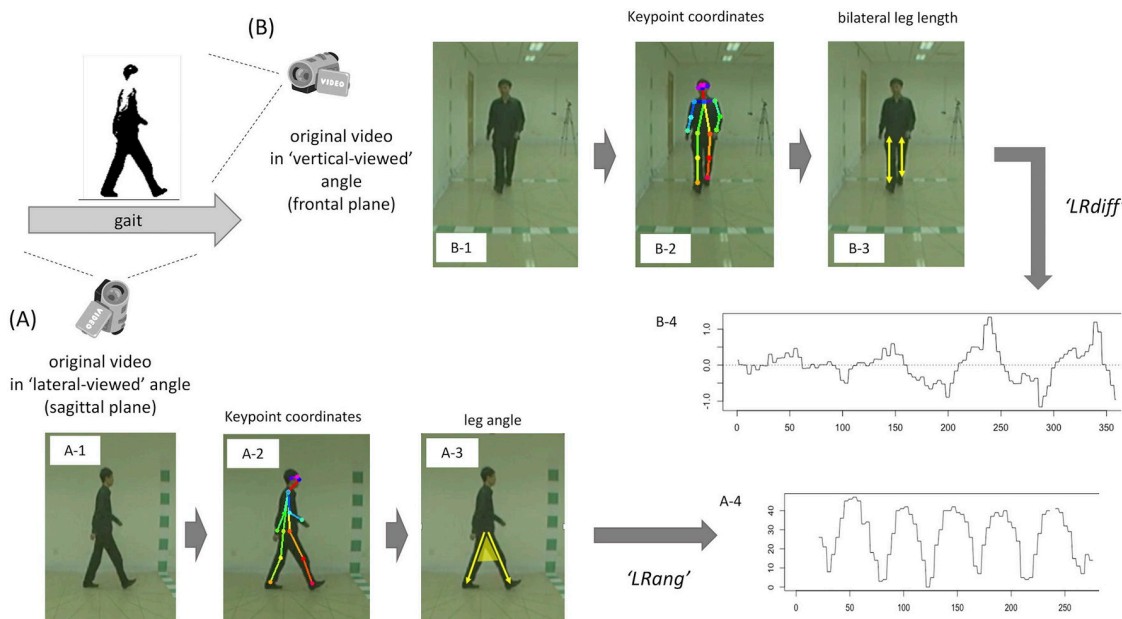

**Fig 2. Detailed feature extraction process using the OpenPose application.** The detailed process corresponds to the upper line in Fig 1. CASIA Dataset-B contains gait sequences of the same gait recorded from different angles simultaneously: (A) laterally viewed angle; and (B) frontally viewed angle. Flow (A-1) to (A-4) shows the process for laterally viewed movies, while flow (B-1) to (B-4) shows that for frontally viewed movies. After estimating keypoints (A-2, B-2), joint coordinate data of each frame are converted to gait features: leg angle (A-3) and difference in bilateral leg-length ratio (B-3). Then sequential waveform data are obtained (A-4, B-4) to calculate gait cycle frequency in the subsequent steps. Note that 1 cycle in the sequential LRdiff (B-3) corresponds to 2 walked steps (= 1 gait cycle), while 1 cycle in the sequential LRang (A-3) corresponds to 1 walked step (= half of 1 gait cycle), so the raw frequency of sequential LRang in laterally viewed movies is multiplied by 0.5 before being converted to a gait frequency.

## 2D keypoint estimation with OpenPose

Then 2D real-time 18-keypoint body/foot estimation was performed on each image by running one of the OpenPose distributions [9] on macOS Mojave version 10.14.3. As shown in the S1 Fig, the keypoints roughly correspond to body parts/joints as follows: 0 = nose, 1 = heart, 2 = right shoulder, 3 = right elbow, 4 = right wrist, 5 = left shoulder, 6 = left elbow, 7 = left wrist, 8 = right hip, 9 = right knee, 10 = right ankle, 11 = left hip, 12 = left knee, 13 = left ankle, 14 = right eye, 15 = right ear, 16 = left eye, and 17 = left ear. In S1 Fig, the keypoints on eyes/ears (nos. 14–17) are omitted from labeling for simplicity. The sequential keypoint coordinate data are thus obtained for each gait sequence (Figs 1C, 2A-2 and 2B-2).

## Feature extraction

As one of the basic gait parameters, here we focused on cadence, which is defined as the number of steps per minute (steps/min). We also use gait frequency (Hz), which is defined as the number of gait cycles–one pair of steps–per second; therefore, a 1-Hz gait frequency is equal to the cadence of 120 steps/min. This focus on cadence is mainly due to the reasons derived from the specification of OpenPose: 2D estimation and unavailability of actual distance information. An earlier spatiotemporal gait analysis in PD versus control showed that parkinsonian gait–associated findings including shorter gait step length, lower cadence (steps/min), shorter stride duration, lower toe clearance and insufficient foot elevation, or more insufficient flexion of the hip and knee joints [16,17]. These features are usually extracted from the 3D motion capture recordings, among which the sagittal plane data are essential. However, the gait videos recorded in the daily clinical setting (e.g. crowded and straightforward hospital hallway) are

often filmed from the frontally viewed angle (i.e. coronal plane of gait) but not from the laterally viewed angle (i.e. sagittal plane of gait). The recordings from the frontally viewed angle would be less informative for identifying the temporal change of joint angles (e.g. knee, hip, ankle), foot progression, foot clearance, or foot contact timing. In addition, since OpenPose does not require scales or calibrations as input, we in turn cannot directly measure the actual distance between the detected keypoints. This means that it is impossible to obtain many spatial gait parameters such as gait speed (m/sec), step length (m), step width (m), or foot clearance (cm) without additional external calibrations.

Originally, the CASIA dataset-B contains gait sequences of n = 124 identical participants, whose age was mostly in their 20s and sex ratio (male:female) was approximately 3:1, and each of whom completes a total of 6 different gait sequences: for each gait sequence, the laterally viewed movie and the frontally viewed movie are recorded simultaneously (but not synchronously). Among them, we used n = 117 pairs of laterally and frontally viewed videos from the identical participants' gait sequences (Table 1) after excluding gait sequence pairs in which there is an apparent discrepancy in the replay speed between laterally and frontally viewed movies. The median length of the frontally viewed movie was 3.8 seconds (interquartile range, 3.3–4.3 sec) and the length of laterally viewed movie was median 3.0 seconds (interquartile range, 2.8–3.2 sec).

As the first step in calculating gait cadence, gait-related features were extracted (Fig 1D). In the laterally viewed movie, we used the time-series data of angle degree between the legs (LRang;

Fig 2A-3 and 2A-4) calculated by the following equation: $LRang = \cos^{-1}\left( \dfrac{\overrightarrow{P_8P_{10}} \cdot \overrightarrow{P_{11}P_{13}}}{(|\overrightarrow{P_8P_{10}}| \cdot |\overrightarrow{P_{11}P_{13}}|)} \right)$

$(0° \leq LRang \leq 180°$, where $P_k$ denotes the 2D positional vector of the $k$th keypoint in S1 Fig: $k = 1, 2, \ldots 16)$. In the frontally viewed movie, we used time-series data of the difference in the left-to-right leg length ratio subtracted from the right-to-left leg length ratio (LRdiff: Fig 2B-3

and 2B-4) calculated by the following equation: $LRdiff = \left( \dfrac{|\overrightarrow{P_{11}P_{13}}|}{|\overrightarrow{P_8P_{10}}|} - \dfrac{|\overrightarrow{P_8P_{10}}|}{|\overrightarrow{P_{11}P_{13}}|} \right)$. These data are

obtained as the time-series waveform data (Fig 2A-4 and 2B-4, respectively). In reference to the gait cycle [18], during the normal-speed gait displayed by healthy control subjects, LRang would become 0° during the midstance or mid-swing period and would peak at around the terminal stance or terminal swing period. LRdiff would become 0° at around the midstance or mid-swing period and reach the maximum/minimum value at around the pre-swing or loading-response period. These raw data are band-pass filtered at the range of 0.1–7 Hz (Fig 1E) in reference to an earlier gait analysis study [19].

## Calculating cadence in normal gait movies

Calculating cadence in each gait sequence now can be considered the problem of pitch detection from the sequential LRdiff waveform data in frontally viewed movies (Fig 2B-4) and from the sequential LRang waveform data in laterally viewed movies (Fig 2A-2). Note that 1 cycle in the sequential LRdiff (Fig 2B-3) corresponds to 2 walked steps (= 1 gait cycle), while the 1 cycle in the sequential LRang (Fig 2A-3) corresponds to 1 walked step (= half of 1 gait cycle), so the raw frequency of sequential LRang in laterally viewed movies is multiplied by 0.5 before being converted to gait frequency.

Then, for the purpose of pitch detection, we used the short-time ACF (ST-ACF) [10, 11], an applied form of STFT [12,13], for normal gait movies from CASIA Dataset-B. Note that we do not use STFT for frequency detection as in acoustic analysis because of the suspected inaccuracy of calculated gait hertz at the bandwidth of the gait analysis. Since the frequency

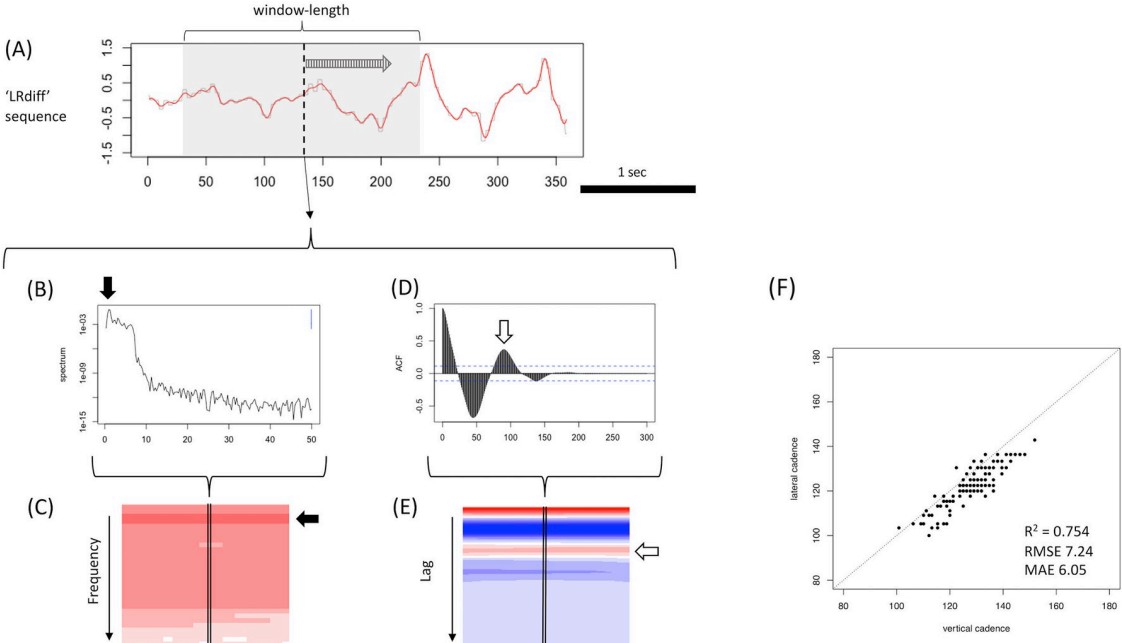

**Fig 3. Short-time ACF used to calculate cadence of a normal gait sequence from CASIA Dataset-B.** (A) The filtered LRdiff sequence of control gait movies from CASIA Dataset-B. For these normal gait movies, we applied ST-ACF with a window length of 2 seconds and a shift-length of 0.01 second (A). In ST-ACF (D), the reciprocal of the lag at the peak of the second positive phase after the initial negative peak (white arrow in D) was selected as the representative frequency. When the sequential ST-ACF matrix is plotted on a heatmap (E), the selected lag is seen as the second from the top and most red horizontal line (white arrow in E). STFT (B, C) is shown as a corresponding example to the ST-ACF: sequential STFT is plotted as a heatmap (C), where the minimum value is white and the maximum color is most-red. The selected frequency just corresponds to the height of the horizontal most-red line (filled arrow in C).

resolution in STFT is the reciprocal of the window length (sec) and the window length would be less than 10 seconds at most in the normal gait movies recorded at actual clinical settings, the frequency resolution in STFT shall be lower than that in ST-ACF at any rate used here. Actually, the median length of the CASIA dataset movies was approximately 3 seconds; therefore, the frequency resolution would be insufficient (e.g. 0.33–0.5 Hz). The window length was 2.0 seconds for normal control gait sequences and 3.0 seconds for pathological gait sequences to include at least 2–3 steps within the short-time window and secure sufficient frequency resolution while balancing with the temporal resolution [11] and the limited length of each movie. The shift length was 0.01 second, which is equal to the reciprocal of frame fps (= 100 here). A Hann function was applied to each window to smooth the bilateral margins [12] (Fig 1F and 1G). Then, we chose the lag at which the autocorrelation value took the largest value in the secondary-positive phase after the initial negative phase (e.g. white arrow in Fig 3D); the reciprocal of the chosen lag time (0.01 sec/lag) corresponds to the representative gait frequency. This is the general criteria we used here to derive cadence using ACF.

The calculated cadence (= 120 × gait frequency) on the same gait sequence was then compared between frontally viewed and laterally viewed movies, and we confirmed their computational consistency using the determinant coefficient ($R^2$), root mean square error (RMSE), and mean absolute error (MAE).

## Calculating cadence in shuffling gait movies

We conducted the same data processing as described above of frontally viewed movies of our two PD patients (Pt. 1 and Pt. 2 in Table 1). These movies were recorded based on the written

informed consent obtained from these patients. These movies were at 30 fps with a resolution of 960 × 540. Since their gait movies were recorded from the frontally viewed angle only, we used the LRdiff as the key gait feature in the following analysis.

For the case (Pt. 2) of abnormal gait with irregular gait rhythm, involuntary leg oscillations, or prominent freezing, the pitch detection framework by ACF as used in normal gait movies sometimes did not work well (we describe this point in the Results section), so we newly introduced metrices to quantify statistical distances between the ST-ACF at each moment and the typical ST-ACF using a dynamic time-warping (DTW) algorithm (package *dtw* in R) [20] (Fig 1H and 1I) and k-nearest neighbor (kNN)–based anomaly detection algorithm (package *FNN* in R) [21,22].

Note that we have not applied the same similarity-based clustering for the raw sequential LRdiff data itself because of the temporal characteristic of the LRdiff waveform. In the gait movies recorded from the frontally viewed angle, the amplitude of the sequential LRdiff gradually increased along with the subjects walking to the camera (Fig 3A; a case without zoom-out); thus, such a skewed waveform pattern as reference can have an undesired influence. The zoom status and height of the video camera can also influence the basic waveform sequence, as vulnerability to these factors will lessen the practical applicability of the currently proposed method. In contrast, the cyclic information converted to ST-ACF would be less vulnerable to such temporal changes or other factors related to filming condition.

As a metric for periodicity, the distribution in DTW- or kNN-based distances is comparable between different gait sequences as long as the same reference and preprocessing configurations are used. We conducted one-way analysis of variance (ANOVA) for the distance metric distribution between different disease status groups (e.g. healthy control for CASIA sequences, mild PD for sequences from Pt. 1, and severe PD with FOG for sequences from Pt. 2). In addition, a ROC analysis for discriminating disease status was also performed using the *pROC* package in R [23].

## Statistical analysis

The data processing/analysis except for the 2D keypoint estimation by OpenPose was performed using R version 3.5.1.

## Ethical

The participants in this manuscript have given written informed consent for this study. This work was conducted in accordance with the ethical standards laid down in the 1964 Helsinki declaration. This study was approved by our hospital's institutional ethics committee (ID: 2339-(3)).

## Results

### Cadence estimation from movies of normal gait

Fig 3 shows the applied results of frequency detection by ST-ACF (Fig 3D and 3E) for normal gait movies from CASIA Dataset-B compared with the corresponding method by STFT (Fig 3B and 3C). The raw sequence (LRdiff or LRang) is plotted as shown in Fig 3A. The window length was determined to be 2.0 seconds for normal control gait sequences (Figs 3 and 4) and 3.0 seconds for pathological gait sequences (Figs 4–6) to include at least 2–3 steps within the short-time window and secure sufficient frequency resolution while balancing the temporal resolution [11] and the limited length of each movie. As on the STFT time-frequency spectrogram (Fig 3C), the derived sequential ST-ACF was plotted on a time-frequency 2D heatmap

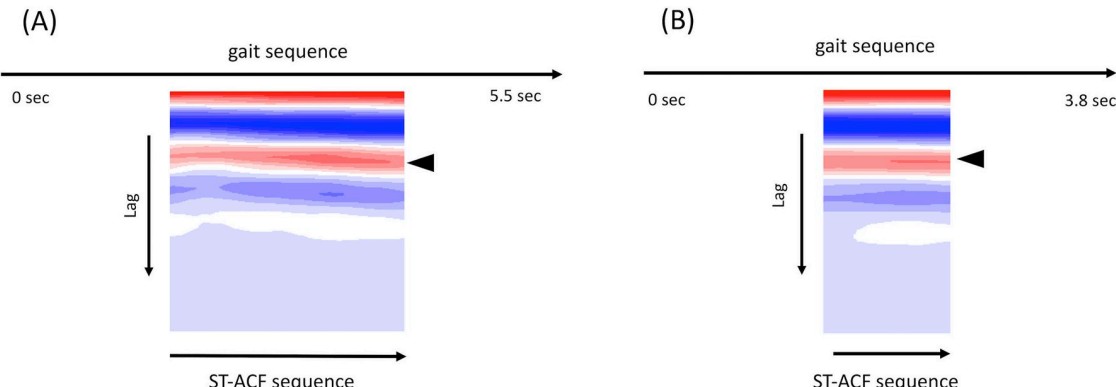

**Fig 4. Applied result of cadence calculation for gait sequences of a mildly affected PD patient.** (A) and (B) show the graphical summary of the sequential ST-ACF obtained from the gait sequences of the same PD patient (Pt. 1) recorded in our hospital's hallway before (A, 5.5 sec) and after (C, 3.8 sec) DBS treatment (frontally viewed movies only). Representative gait frequency is calculated as the reciprocal of the median lag at the second positive peak in each moment ST-ACF (= most red color on the heatmaps) (arrowheads in A, B). The calculated gait cadence was 140.5 steps/min before DBS and 134.1 steps/min after DBS.

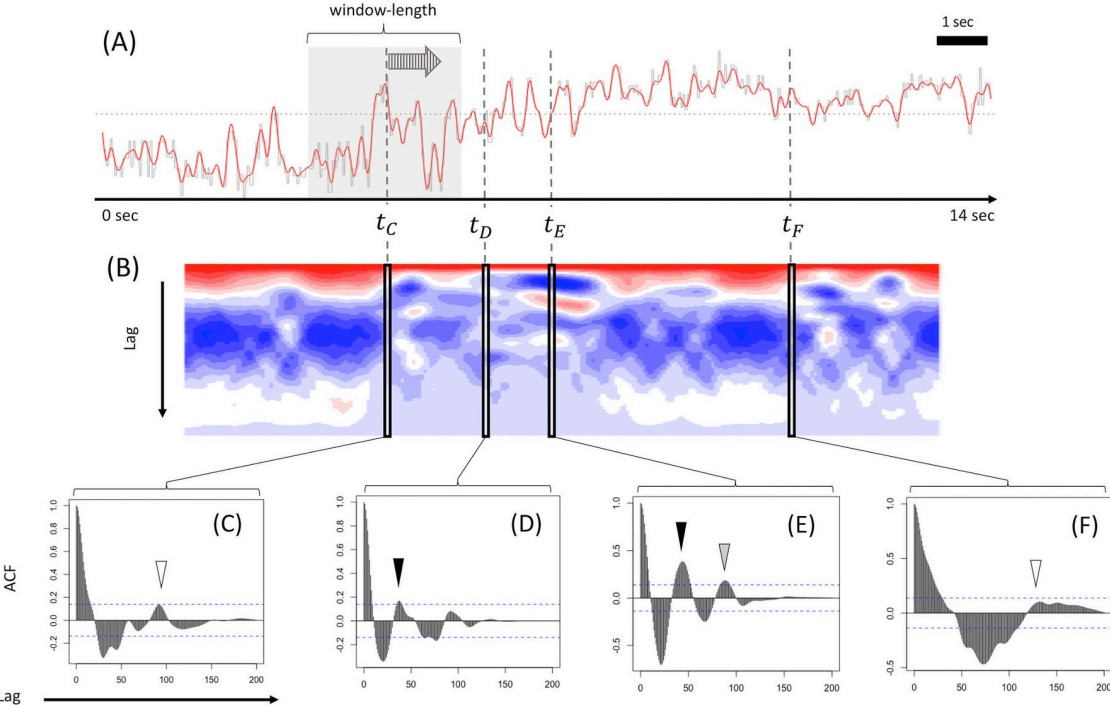

**Fig 5. Example of cadence calculation for a significantly abnormal gait sequence.** (A) Filtered sequential 'LRdiff' of gait sequence of our PD patient's (Pt. 2), whose gait mixed with significant FOG and small steps. The movie is 14 seconds long, the time she took to walk less than 2 m. Many instances of freezing and involuntary oscillations were observed during the gait. The sequential ST-ACF of 11 seconds' duration (subtracting window length of 3 seconds from the raw movie length) obtained from (A) is shown as (B). (C)–(F) show the ST-ACF at the corresponding $t_C$ – $t_F$ timings in (A, B). (C)–(F) demonstrate examples in which accurate gait frequency detection is confirmed (D, E) or doubtful (C, F) when applying the same procedure used for the normal gait movies as shown in Fig 3.

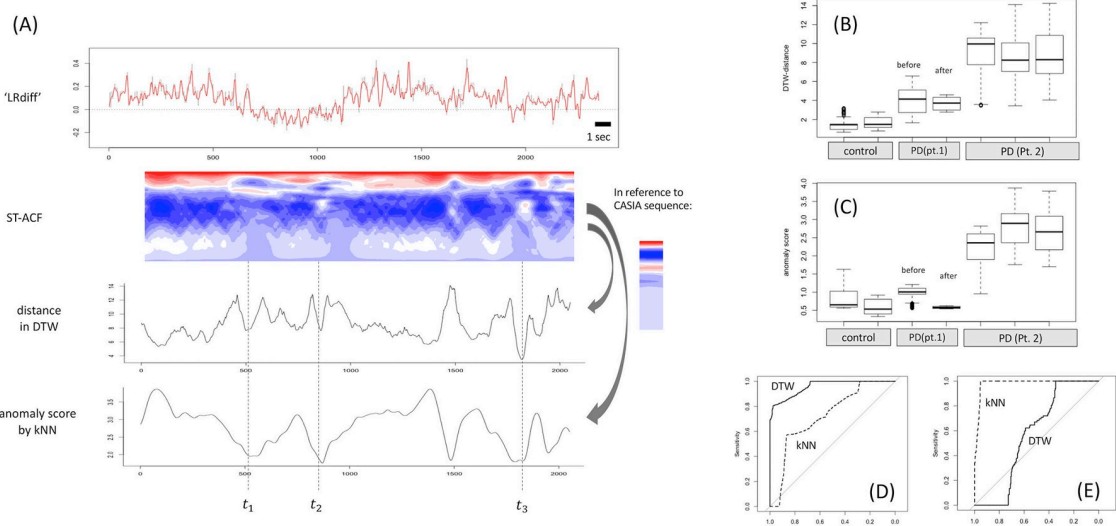

**Fig 6. Example of identifying periodic gait steps among the mixed FOG and metrics comparison between different gait sequences.** (A) Data obtained from the same patient (Pt. 2) as in Fig 5. In reference to the ST-ACF of control gait that is randomly selected from CASIA Dataset-B sequences, the similarity of ST-ACF at each moment to that of normal gait is measured in terms of DTW-based distance (second from bottom row in A) or kNN-based anomaly (bottom row in A). As long as the same reference and preprocessing configurations are used, the DTW- or kNN-based distance are comparable between different gait sequences as summarized in boxplots (B) for DTW and in boxplots (C) for kNN. In (B) and (C), there was a significant difference in statistical distance/anomaly value among the 3 subgroups: control gait, affected gait from Pt. 1 (Fig 4), and severely affected gait from Pt. 2 (one-way analysis of variance; $p < 2.2e^{-16}$). Regarding the difference between control gait sequence and gait from Pt. 1 (Fig 6D), discrimination performance was better for DTW (area under the curve [AUC], 0.957) than for kNN (AUC, 0.754) ($p < 2.2e^{-16}$ for DeLong's test). Meanwhile, when comparing gait sequences before versus after DBS treatment from Pt. 1 (Fig 4A and 4C), discrimination performance was better for kNN-based distance (AUC, 0.980) than for DTW-based distance (AUC, 0.572) ($p < 2.2e^{-16}$ on DeLong's test) (Fig 6E).

(blue, minimum value; red, maximum value) (Fig 3E). The selected representative frequency can be found on the heatmap as the height value of the second-top horizontal red band (white arrow in Fig 3E).

We then compared consistency of the cadence between the laterally viewed movie and the frontally viewed movie of the same gait. The result is plotted in Fig 3F, which shows good consistency between them: $R^2 = 0.754$, RMSE = 7.24, and MAE = 6.05 (steps/min).

## Cadence estimation from movies of mildly affected gait from our PD patients

Next, we applied the same procedure to actual clinical movies of our PD patients recorded in the frontally viewed angle filmed in our hospital's hallway (Pt. 1 and Pt. 2 in Table 1): one patient (Pt. 1) was a female PD patient in her 60s who showed a clear On-Off phenomenon and corresponds to Yahr 3 during the Off state. She was recorded for her gait with mild small steps before deep brain stimulation (DBS) implementation; after DBS, her gait had improved. The other patient (Pt. 2) was a female PD Yahr 4 patient in her 70s who was suffering from prominent FOG in daily life with a high total FOG questionnaire score (17 or higher). She was filmed to capture her significantly shuffling gait.

Fig 4A shows the sequential ST-ACF heatmap as of Fig 3E for gait sequences of Pt. 1 during On period before DBS treatment, recorded in our hospital's hallway (movie length: 5.5 sec). She showed mild small steps with a cadence of 140.5 steps/min (Fig 4A). After DBS treatment

(movie length: 3.8 sec), her gait became almost normal with an improved foot clearance and a cadence that had decreased slightly to 134.1 steps/min (Fig 4B).

## Cadence estimation from movies of severely affected gait from our PD patients

Unlike in the case of Pt. 1, whose gait was generally periodic, in Pt. 2, the gait was clearly abnormal and the pitch detection framework by ACF as used in normal gait movies (Figs 3 and 4) did not always work well. We explain this problem in Fig 5, which shows a graphical summary of one of the gait sequences of Pt. 2 in which she took 14 seconds to walk less than 2 m (unfortunately, we cannot provide the gait movie itself due to the limitation posed by the patient's informed consent). She had a small step and small arm swing with a forward-bent posture and froze frequently while walking. Fig 5A shows the sequential 'LRdiff' of the gait, while its sequential ST-ACF with a shift length of 3.0 seconds is shown in the heatmap in Fig 5B. The transient periodic steps are observed on the movie at around the timing $t_D$, and the gait frequency at that moment is calculated in Fig 5D as the reciprocal of the lag at a black arrowhead, although the height of the second positive peak after the initial negative phase was barely over the 95% confidential interval level. However, at timing $t_C$ or $t_F$ at ST-ACF (Fig 5C and 5F), the highest peak of the second positive phase (shown with white arrowheads) after the initial negative peak had insufficient height (less than 95% confidence interval as shown with a dotted line), making it impossible to recognize the original short-time LRdiff sequence as having significant periodicity. In addition, at timing $t_E$ at ST-ACF (Fig 5E), there is another peak candidate (white arrowhead) next to the peak of the second positive phase, and there is some room to select the lag at the latter peak as such in expected cases where the latter peak is significantly higher than the former one. In other words, it can sometimes be annoying to determine a single lag value for a frequency calculation.

These uncertainties in determining a single representative frequency are ultimately inevitable limitations in this short-time window approach, so instead of trying to determine a single lag to calculate gait frequency within each short-time window, we handled the ACF distribution from the point of the sequential ST-ACF matrix. Since the ST-ACF is the summary of sequential information in terms of periodicity, it is possible to interpret the ST-ACF as the representation for the degree of rhythmic gait within the selected window period. If we can parameterize to what degree the ST-ACF pattern at each timing is close to that of a normal gait, it is expected that we can identify periodic gait steps–whether a normal gait or periodic small step gait–even in the midst of plenty of noise due to irregular leg movements. This is why we used a DTW or kNN algorithm to quantify statistical distances between the ST-ACF at each moment and the typical ST-ACF. The DTW is a time-series clustering algorithm that calculates the statistically significant distance of two waveforms' sequential patterns (= 0 between the same sequences) and has advantage over other types of sequential clustering methods in that it can compare two waveform data of which sequential length or sequential phases are significantly different. Therefore, use of the DTW is appropriate when measuring similarity in the sequential pattern of ST-ACF with varying sequential lengths or varying lags at the second positive peak. By using the normal control gait from CASIA dataset as the ST-ACF waveform reference, we can identify the timing at which temporal gait is similar to the control gait. In addition, kNN-based anomaly detection is a simple and frequently used algorithm for sequential anomaly events such as paroxysmal arrhythmia on electrocardiography [22].

An applied result of DTW- or kNN-based distances is shown in Fig 6A, in which the gait sequence differs from that in Fig 5 but is obtained from the same patient (Pt. 2). In this gait

sequence, the patient took as long as 23 seconds to walk less than 5 m. The plot on the second-bottom row in Fig 6A is a serial DTW-based distance in reference to the randomly selected control gait sequence from CASIA Dataset-B with a window length of 3 sec. The bottom-most row plots the serial kNN-based distance. The cadence of the gait sequence shall be calculated reliably for the timings at which the DTW and/or kNN distance reaches the outstanding negative peak as shown with a vertical dotted line ($t_1$, $t_2$, and $t_3$). The calculated cadences were 179.1 steps/min at $t_1$, 118.8 steps/min at $t_2$, and 104.3 steps/min at $t_3$ (Fig 6A).

These sequential distances were then summarized to compare the different gait sequences. As shown in boxplots for DTW distance (Fig 6B) and kNN distance (Fig 6C), there was a significant difference in their distance distribution among the 3 subgroups of gait sequences: control gait (from CASIA dataset), mildly to moderately affected gait by PD (from Pt. 1; Fig 4), and severely affected gait by FOG from Pt. 2 (one-way ANOVA; $p < 2.2e^{-16}$). The difference in values between the severely affected gait (from Pt. 2) and the other gait sequences (from control or Pt. 1) is apparent. When discriminating gait sequence of the control subjects from that of Pt. 1, discrimination performance was better in the DTW-based distance (area under the ROC curve [AUC], 0.957) than in kNN-based distance (AUC, 0.754) ($p < 2.2e^{-16}$ on DeLong's test) (Fig 6D). Meanwhile, when comparing gait sequences from Pt. 1 before and after DBS treatment (Fig 4A and 4B), discrimination performance was better in the kNN-based distance (AUC, 0.980) than in the DTW-based distance (AUC, 0.572) ($p < 2.2e^{-16}$ on DeLong's test) (Fig 6E).

## Discussion

Gait movies are occasionally recorded in daily clinical practice of patients with neurological diseases, but there is a barrier for these movies to be quantitatively analyzed because of the requirement for specific gait analysis devices. To overcome this barrier, here we used Open-Pose to extract gait parameters from gait movies recorded by a monocular home video camera filmed from the frontally viewed angle. As a result, we calculated the gait cadences of normal gait sequences with fair computational consistency between the laterally and frontally viewed movies using the short-time pitch-detection approach. Furthermore, we also demonstrated an example of PD patient movies filmed in actual in-hospital clinical settings as well as how to identify periodic gait steps among abnormal gait sequences with prominent FOG and extract their cadence using DTW- and kNN-based distance from the ACF pattern of a normal gait. Our proposed method shows that gait cadence and its time-serial change can be visualized and parameterized with fair reproducibility, even from gait videos recorded with a monocular home video camera.

Our proposed method consists of two major characteristics: first, extracting sequential gait features (e.g. LRdiff in the frontally viewed movie and LRang in the laterally viewed movie) via body joint coordinates estimation using OpenPose to calculate cadence; and second, quantifying gait steps' periodicity to improve the reliability of the calculated cadence. Regarding the first characteristic, the current OpenPose-based gait analysis has great potential due to its wide applicability regardless of some possible limitations: for example, as mentioned in the Methods, the OpenPose only estimates 2D joint coordinates in each frame and depth information is not available, so the accurate step progression or foot clearance are not obtainable due to parallax error in laterally viewed movies [24] or perspective error in frontally viewed movies [25], which is one of the major reasons why a 3D but not a 2D video capture system is now widely used. In addition, gait parameters including actual length measurements (e.g. step width or gait speed) are not available. Despite these limitations, as long as we focus on the gait periodicity including cadence to extract as gait features, the proposed OpenPose-based method works

effectively even for gait movies recorded in clinical settings at hospital, mostly from the frontally viewed angle.

Regarding the second major characteristic of the current study, we used ST-ACF instead of STFT to detect periodic gait steps because the latter has poor frequency resolution at the bandwidth of gait analysis (maximum 10 Hz here). By quantifying the sequential ST-ACF pattern in reference to that of normal gait in a data-driven manner, we could identify the timing at which gait steps were distinguishable from the concurrent FOG and involuntary leg oscillations. Since it is sometimes difficult to accurately annotate step timing for the gait sequence with severe small steps mixed with prominent FOG or involuntary leg oscillations, our data-driven approach can complement such difficulties. In addition, our approach could visualize the temporal distribution of gait steps with varying degrees of periodicity.

The summarized sequential distance metrics (Fig 6B and 6C) showed a clear proportion to the subjects' disease status. Compared to the apparently large statistical distance in gait sequence from PD with severe FOG, the difference in the sequential statistical distance between the gait of mild PD and the control gait was relatively small. Such a difference seemed to coincide with the difference in their gait by visual appearance. Although the DTW-based distance showed better ability than the kNN-based distance to discriminate control gait from mild PD gait, it should be evaluated with larger samples to determine which distance algorithm better represents the regular periodicity of gait sequence.

In the field of acoustic analysis, in which one separates the source from the mixed sound or voice, the application of nonnegative matrix factorization (NMF) to the STFT spectrogram is frequently used [26]; however, we did not use NMF in this study to quantify or classify ST-ACF patterns. While the factorized basis matrix can be considered the representative ACF pattern and the activation matrix can be considered the sequential distribution of each principle basic ACF pattern, the applicability of NMF to ST-ACF seems doubtful. In other words, it is uncertain whether we can assume that ACF waveform is linearly additive. This is why we used DTW- or kNN-based quantification of ST-ACF patterns at each moment instead of NMF.

Our study has some additional limitations that require consideration. First, there is currently no clear cut-off threshold for DTW- or kNN-based distance, and our results depend on the kind of gait movies used as reference. Second, it was not ascertained whether the summarized sequential statistical distance metrics reflect the features of pathological gait. Thus, the standardization of ACF pattern measurements would be essential for the practical use of this approach. Third, the number of patients, and their PD clinical stages, and the included movies are very small in this study, making the results statistically less robust ones and lessening the generalizability of our methods.

Furthermore, the current ACF pattern–based gait quantification may not always be robust; in the current data-driven manner alone, a mere rhythmic step on the start line without a walking start cannot be distinguished from the small step gait at a similar step speed. In other words, our current approach can only identify periodic gait steps, not foot progression itself. The length of ACF window is another source of limitation. We used 3 seconds of window for movies from our PD patient as a result of trade-off between the number of eligible video sequences and the temporal/frequency resolution in the pitch detection; however, this window length may actually be insufficient for progressed PD patients with slower gait speed to include 2–3 of gait steps within the window, which may eventually lead to the decreased frequency resolution.

In addition, the applicability of the proposed metric for disordered gait due to other neurological diseases with gait disturbances such as spinocerebellar ataxia, normal pressure hydrocephalus, progressive supranuclear palsy, spastic paraparesis, or stroke sequelae, was not

proven. Developing other metrics to complement these weak points is required to increase the applicability of our method to more complex gait situations. Metrics representing horizontal hip swing, or left-to-right difference in the distribution of ankle/knee/hip joints' 2D velocity vector after adjustment with the suspected deviation in the direction between the camera and gait, may be potential candidates to characterize the features of abnormal gait in these neurological diseases.

Future studies are needed to secure gait movies of sufficient sample size and movie length in which gait status is largely varied. There may be a limit to parameterizing salient and robust gait features that can be applied to any types of gait movies under the assumption of extracting them using OpenPose. It may be helpful to use sequential deep learning to discriminate gait status without clearly defining gait features. Furthermore, if we can obtain spatiotemporal parameters by simultaneously using an accelerometer or force plate while recording gait movies, spatiotemporal parameter values may conversely regress.

In conclusion, here we demonstrated how to identify an effective gait and calculate its cadence in normal and pathological gaits from 2D movies recorded with a home video camera. The proposed method may broaden the range of movies for which gait analyses can be conducted.

## Supporting information

**S1 Fig. Scheme of 2D keypoint coordinates.** The 16 keypoints' coordinates are estimated independently by OpenPose for each frame. The keypoints represented are as follows: 0 = nose, 1 = heart, 2 = right shoulder, 3 = right elbow, 4 = right wrist, 5 = left shoulder, 6 = left elbow, 7 = left wrist, 8 = right hip, 9 = right knee, 10 = right ankle, 11 = left hip, 12 = left knee, 13 = left ankle, 14 = right eye, 15 = right ear, 16 = left eye, and 17 = left ear. The keypoints on eyes or ears (nos. 14–17) are omitted from the labeling for simplicity.
(TIFF)

## Author Contributions

**Conceptualization:** Kenichiro Sato, Yu Nagashima.

**Data curation:** Kenichiro Sato.

**Formal analysis:** Kenichiro Sato.

**Investigation:** Kenichiro Sato, Yu Nagashima.

**Methodology:** Kenichiro Sato.

**Project administration:** Kenichiro Sato.

**Software:** Kenichiro Sato.

**Supervision:** Tatsuo Mano, Atsushi Iwata, Tatsushi Toda.

**Validation:** Yu Nagashima.

**Writing – original draft:** Kenichiro Sato.

**Writing – review & editing:** Kenichiro Sato, Tatsuo Mano, Atsushi Iwata.

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
