## [Decision Letter · Decision Letter 0]

17 Oct 2019

PONE-D-19-26601

Quantifying normal and parkinsonian gait features from home movies: Practical application of a deep learning–based 2D pose estimator

PLOS ONE

Dear Dr. Iwata,

Thank you for submitting your manuscript to PLOS ONE. After careful consideration, we feel that it has merit but does not fully meet PLOS ONE’s publication criteria as it currently stands. Therefore, we invite you to submit a revised version of the manuscript that addresses the points raised during the review process.

We would appreciate receiving your revised manuscript by Dec 01 2019 11:59PM. To enhance the reproducibility of your results, we recommend that if applicable you deposit your laboratory protocols in protocols.io, where a protocol can be assigned its own identifier (DOI) such that it can be cited independently in the future. For instructions see: http://journals.plos.org/plosone/s/submission-guidelines#loc-laboratory-protocols

We look forward to receiving your revised manuscript.

Kind regards,

Ken Arai

Academic Editor

PLOS ONE

Journal Requirements:

2. Please provide additional details regarding participant consent. In the ethics statement in the Methods and online submission information, please ensure that you have specified (1) whether consent was informed and (2) what type you obtained (for instance, written or verbal, and if verbal, how it was documented and witnessed). If the need for consent was waived by the ethics committee, please include this information.

3. We note that Figures 2, 4 and 6 includes an image of a [patient / participant / in the study]. 

Reviewers' comments:

Reviewer's Responses to Questions

**Comments to the Author**

1. Is the manuscript technically sound, and do the data support the conclusions?

Reviewer #1: Yes

Reviewer #2: Yes

2. Has the statistical analysis been performed appropriately and rigorously? 

Reviewer #1: Yes

Reviewer #2: I Don't Know

3. Have the authors made all data underlying the findings in their manuscript fully available?

Reviewer #1: Yes

Reviewer #2: Yes

4. Is the manuscript presented in an intelligible fashion and written in standard English?

Reviewer #1: Yes

Reviewer #2: Yes

5. Review Comments to the Author

Reviewer #1: Sato and his colleagues present an important work using OpenPose. This is an interesting study that may become a standard method for evaluating the gait disturbance. I have noted several minor corrections as below. I would be glad if my comments could help your manuscript.

1. This study assessed patients with PD. I think this method can be widely applicable other neurological patients who show the gait disturbance, not only cerebella ataxia, but also normal pressure hydrocephalus, spastic paraparesis, and more. If you can show the future prospects, please describe.

2. About the recorded plane of gait, CASIA evaluated sagittal view. If you evaluated sagittal view in patients with PD, please provide.

3. Please show the annotation of the horizontal axis in Figure 4 (B) and (D).

Reviewer #2: The manuscript by Sato et al entitled ‘Quantifying normal and parkinsonian gait features from home movies: Practical application of a deep learning–based 2D pose estimator’ proposed the utility of 2D movies recorded by home video camera for identification of normal and pathological gaits. The authors obtained gait features of normal control (from CASIA database) and PD patients in their hospital by using OpenPose, and estimated cadence of gait steps from the sequential gait features. Their analyzes showed that the quantified periodicity of each gait sequence corresponded with disease status. Thus, they concluded that their method might be useful to analyze gait movies, which have been underutilized.

Their data seems to be preliminary, however, the concept is very interesting and has a good point to make. Following points can be considered to be added.

1) As the authors already addressed in Page12 in Method section, the window length of gait movies is insufficient especially for PD patients. This point should be expanded/discussed with presumable drawbacks, and addressed in the limitation part in Discussion section.

2) Other filming conditions, such as distance between the patient and camera or focal length etc, should be listed.

3) Limited number and severity range (Yahr scale) of PD patients were analyzed in this study.

6. PLOS authors have the option to publish the peer review history of their article (what does this mean?). If published, this will include your full peer review and any attached files.

Reviewer #1: No

Reviewer #2: No

---

## [Author Response · Author response to Decision Letter 0]

24 Oct 2019

2. Please provide additional details regarding participant consent.

(Reply) As per your comment, we described that we have obtained written informed consent to record films, analyze, and report as the research articles.

3. We note that Figures 2, 4 and 6 includes an image of a [patient / participant / in the study]. As per the PLOS ONE policy (http://journals.plos.org/plosone/s/submission-guidelines#loc-human-subjects-research) on papers that include identifying, or potentially identifying, information, the individual(s) or parent(s)/guardian(s) must be informed of the terms of the PLOS open-access (CC-BY) license and provide specific permission for publication of these details under the terms of this license. Please download the Consent Form for Publication in a PLOS Journal (http://journals.plos.org/plosone/s/file?id=8ce6/plos-consent-form-english.pdf). The signed consent form should not be submitted with the manuscript, but should be securely filed in the individual's case notes. Please amend the methods section and ethics statement of the manuscript to explicitly state that the patient/participant has provided consent for publication: “The individual in this manuscript has given written informed consent (as outlined in PLOS consent form) to publish these case details”.

(Reply) 

Instead of obtaining the “signed informed consent as outlined in PLOS form”, we deleted the patients’ images on Figure 4 and Figure 5, and we also revised the manuscript and figure legends. Other patients’ information such as decade-in-age or their disease stage are not detailed information and therefore they are not identifiable data.

 As to the Figure 2, the images were obtained from the CASIA database which is an open database available from their website by anyone, and in its strict sense they are not the actual participants of our study. The database terms of use do not prohibit the present way of including images into a research article as in our Figure 2. By these reasons, we believe the Figure 2 would be allowed as it is, without need to remove it.

REVIEWER 1 

1. This study assessed patients with PD. I think this method can be widely applicable other neurological patients who show the gait disturbance, not only cerebella ataxia, but also normal pressure hydrocephalus, spastic paraparesis, and more. If you can show the future prospects, please describe.

 (Reply) We thank for your important comment. As per your comment, we additionally described about other diseases this study is potentially applicable, and the potential future solutions to parameterize the gait abnormalities seen in these diseases as follows:

• In addition, the applicability of the proposed metric for disordered gait due to other neurological diseases with gait disturbances such as spinocerebellar ataxia, normal pressure hydrocephalus, progressive supranuclear palsy, spastic paraparesis, or stroke sequelae, was not proven. Developing other metrics to complement these weak points is required to increase the applicability of our method to more complex gait situations. Metrics representing horizontal hip swing, or left-to-right difference in the distribution of ankle/knee/hip joints’ 2D velocity vector after adjustment with the suspected deviation in the direction between the camera and gait, may be potential candidates to characterize the features of abnormal gait in these neurological diseases. (Line 12-19 on Page 27)

2. About the recorded plane of gait, CASIA evaluated sagittal view. If you evaluated sagittal view in patients with PD, please provide.

 (Reply) We thank for your comment. Unfortunately, we have no laterally-viewed movie for our patients. This is because the movies in our hospital are recorded in the narrow hallways.

3. Please show the annotation of the horizontal axis in Figure 4 (B) and (D).

 (Reply) As per your comment, we added the annotation on Figure 4. 

REVIEWER 2

1) As the authors already addressed in Page12 in Method section, the window length of gait movies is insufficient especially for PD patients. This point should be expanded/discussed with presumable drawbacks, and addressed in the limitation part in Discussion section.

 (Reply) As you indicate, the window length we applied for movies from PD patients were 3 seconds, being relatively short to assure 2-3 steps included within that window. This was trade-off between the number of eligible video sequences and the temporal resolution in the cadence estimation. 

As per your comment, we described the discussion as to this point the Discussion section, as follows: 

• Furthermore, the current ACF pattern–based gait quantification may not always be robust; in the current data-driven manner alone, a mere rhythmic step on the start line without a walking start cannot be distinguished from the small step gait at a similar step speed. In other words, our current approach can only identify periodic gait steps, not foot progression itself. The length of ACF window is another source of limitation. We used 3 seconds of window for movies from our PD patient as a result of trade-off between the number of eligible video sequences and the temporal/frequency resolution in the pitch detection; however, this window length may actually be insufficient for progressed PD patients with slower gait speed to include 2-3 of gait steps within the window, which may eventually lead to the decreased frequency resolution. (Line 3-11 on Page 27)

2) Other filming conditions, such as distance between the patient and camera or focal length etc, should be listed.

(Reply) As per your comment, we additionally described other recording conditions on the Table 1 as far as we know, including the used video model and the distance between the participants and camera. We could not obtain information about the focal length from both of movies of CASIA dataset and our patients, and the former is due to the data missing, and the latter is due to the use of zoom-in/out during recording. 

3) Limited number and severity range (Yahr scale) of PD patients were analyzed in this study.

 (Reply) As you point out, this is actually the limitation of this study. We mentioned about this point as one of the limitations in the Discussion section, as follows:

• Our study has some additional limitations that require consideration. First, there is currently no clear cut-off threshold for DTW- or kNN-based distance, and our results depend on the kind of gait movies used as reference. Second, it was not ascertained whether the summarized sequential statistical distance metrics reflect the features of pathological gait. Thus, the standardization of ACF pattern measurements would be essential for the practical use of this approach. Third, the number of patients, and their PD clinical stages, and the included movies are very small in this study, making the results statistically less robust ones and lessening the generalizability of our methods. (Line 15-19 of Page 26, and Line 1-2 of Page 27)

---

## [Decision Letter · Decision Letter 1]

30 Oct 2019

Quantifying normal and parkinsonian gait features from home movies: Practical application of a deep learning–based 2D pose estimator

PONE-D-19-26601R1

Dear Dr. Iwata,

We are pleased to inform you that your manuscript has been judged scientifically suitable for publication and will be formally accepted for publication once it complies with all outstanding technical requirements.

With kind regards,

Ken Arai

Academic Editor

PLOS ONE

Additional Editor Comments (optional):

Reviewers' comments:

Reviewer's Responses to Questions

**Comments to the Author**

1. If the authors have adequately addressed your comments raised in a previous round of review and you feel that this manuscript is now acceptable for publication, you may indicate that here to bypass the “Comments to the Author” section, enter your conflict of interest statement in the “Confidential to Editor” section, and submit your "Accept" recommendation.

Reviewer #1: All comments have been addressed

Reviewer #2: All comments have been addressed

2. Is the manuscript technically sound, and do the data support the conclusions?

Reviewer #1: Yes

Reviewer #2: Yes

3. Has the statistical analysis been performed appropriately and rigorously? 

Reviewer #1: (No Response)

Reviewer #2: I Don't Know

4. Have the authors made all data underlying the findings in their manuscript fully available?

Reviewer #1: Yes

Reviewer #2: Yes

5. Is the manuscript presented in an intelligible fashion and written in standard English?

Reviewer #1: Yes

Reviewer #2: Yes

6. Review Comments to the Author

Reviewer #1: (No Response)

Reviewer #2: In this revised version of the manuscript entitled ‘Quantifying normal and parkinsonian gait features from home movies: Practical application of a deep learning–based 2D pose estimator’, all required comments and discussions are sufficiently addressed.

7. PLOS authors have the option to publish the peer review history of their article (what does this mean?). If published, this will include your full peer review and any attached files.

Reviewer #1: No

Reviewer #2: No

---

## [Editor Report · Acceptance letter]

6 Nov 2019

PONE-D-19-26601R1 

Quantifying normal and parkinsonian gait features from home movies: Practical application of a deep learning–based 2D pose estimator 

Dear Dr. Iwata:

I am pleased to inform you that your manuscript has been deemed suitable for publication in PLOS ONE. Congratulations! Your manuscript is now with our production department. 

With kind regards,

on behalf of

Dr. Ken Arai 

Academic Editor

PLOS ONE